# CryoEM and stability analysis of virus-like particles of potyvirus and ipomovirus infecting a common host

Ornela Chase[1], Abid Javed[2], Matthew J. Byrne[2,4], Eva C. Thuenemann[3], George P. Lomonossoff [3], Neil A. Ranson [2] & Juan José López-Moya [1✉]

*Sweet potato feathery mottle virus* (SPFMV) and *Sweet potato mild mottle virus* (SPMMV) are members of the genera *Potyvirus* and *Ipomovirus*, family *Potyviridae*, sharing *Ipomoea batatas* as common host, but transmitted, respectively, by aphids and whiteflies. Virions of family members consist of flexuous rods with multiple copies of a single coat protein (CP) surrounding the RNA genome. Here we report the generation of virus-like particles (VLPs) by transient expression of the CPs of SPFMV and SPMMV in the presence of a replicating RNA in *Nicotiana benthamiana*. Analysis of the purified VLPs by cryo-electron microscopy, gave structures with resolutions of 2.6 and 3.0 Å, respectively, showing a similar left-handed helical arrangement of 8.8 CP subunits per turn with the C-terminus at the inner surface and a binding pocket for the encapsidated ssRNA. Despite their similar architecture, thermal stability studies reveal that SPMMV VLPs are more stable than those of SPFMV.

[1] Centre for Research in Agricultural Genomics (CRAG, CSIC-IRTA-UAB-UB), 08193Cerdanyola del Vallès, Barcelona, Spain. [2] Astbury Centre for Structural Molecular Biology, School of Molecular and Cellular Biology, Faculty of Biological Sciences, University of Leeds, Leeds, UK. [3] Department of Biochemistry and Metabolism, John Innes Centre, Norwich Research Park, Norwich NR4 7UH, UK. [4]Present address: Electron Bio-Imaging Centre, Diamond Light Source, Harwell Science and Innovation Campus, Fermi Ave, Didcot, Oxfordshire OX11 0DE, UK. ✉email: juanjose.lopez@cragenomica.es

The *Potyviridae* is the largest family of plant RNA viruses, collectively known as potyvirids, encompassing nearly one third of all known plant viruses. They are distributed worldwide and infect many plants of agricultural importance[1,2]. Currently the family includes 12 genera, members of the genus *Potyvirus* being most abundant, containing over 195 species[3]. The genome of most potyviruses consists of a positive-sense single-stranded RNA of ~10 kb that encodes a large polyprotein that gives rise to different mature gene products, named from N- to C-terminus: P1, HCPro, P3, 6K1, CI, 6K2, NIa-VPg, NIa-Pro, NIb, and CP. Processing of the polyprotein is based on the proteolytic activity of three virus-encoded proteases: P1 and HCPro, acting in cis, and NIa-Pro, acting both in cis and trans[4,5]. In addition, the genomes of all members of the family encode a partially out-of-frame P3N-PIPO gene product, that is translated from genome variants with extra nucleotides generated after a polymerase slippage mechanism occurring in a conserved G2A6 nucleotide motif[6–8]. An equivalent mechanism also serves to produce P1N-PISPO in certain potyviruses infecting sweet potato (*Ipomoea batatas*) plants[9,10]. The genome organization is similarly shared by members of the additional genera in the family, with a few divergences, mostly mapping in the 5' region[11].

A common feature for all potyvirids is the structure of virions as non-enveloped, flexible rod-shaped particles, composed of multiple copies of a unique coat protein (CP) helically arranged around the RNA genome[12,13]. Virions play essential roles in the virus replication cycle and especially during the transmission process. Potyviruses are transmitted by aphid vectors in a non-persistent manner with the participation of CP and HCPro[14–17], while other genera within the family include viruses grouped by distinctive biological characteristics, mainly the different vectors used for transmission, such as the whitefly-transmitted ipomoviruses[18], that are comparatively less studied in their transmission. The classification into genera also takes into account other peculiarities in genome structure and phylogenetic relationships. Evolutionary studies have suggested that the expansion of species of potyviruses might have coincided roughly with the dawn of agriculture, a notion that is further supported by their collective ability to infect a wide range of hosts, and their global presence[19,20].

Virion structures have been determined for several potyviruses, although only three of them, *Watermelon mosaic virus* (WMV), *Turnip mosaic virus* (TuMV) and *Potato virus Y* (PVY), have been resolved by CryoEM[21–23]. By contrast, comparatively few members of other genera within the family have been studied[24,25] and the structures of virions of some genera, including ipomoviruses, are unknown. To compare the virion structures of viruses belonging to different genera, we selected the potyvirus *Sweet potato feathery mottle virus* (SPFMV) and the ipomovirus *Sweet potato mild mottle virus* (SPMMV) as these naturally infect the same host plant, sweet potato. They are both important pathogens: SPFMV is considered the most prevalent viral pathogen of sweet potato worldwide, and although it is often not particularly damaging as a single infection, it can be quite severe in co-infections with *Sweet potato chlorotic stunt virus* (SPCSV, genus *Crinivirus*, family *Closteroviridae*), leading to dramatically high yield losses up to 80%[26,27]. Similarly, SPMMV[28] can infect the same crop and also cause significant yield losses in mixed infections with other viruses like SPCSV[29,30].

As mentioned above, these two viruses belong to genera that include members transmitted by different homopteran insect vectors: aphids in the case of potyviruses and whiteflies in the case of ipomoviruses. The mechanism of aphid transmission in potyviruses has been extensively studied, including the characterization of HCPro as an auxiliary factor essential for the process along with the CP[16]. By contrast, for ipomoviruses there is little information about the role of virions for vector specificity, nor about the possible dependence of transmission auxiliary factors. Interestingly, the HCPro region is absent in ipomoviruses, except in SPMMV, that have instead either single or duplicated P1s[31,32]. While the HCPro is the canonical RNA silencing suppressor (RSS) in most potyviruses[33,34], this function appears to be shifted to P1 in ipomoviruses[35], and even in the case of SPMMV where HCPro is present, P1 was shown to be the product acting as an RSS[36]. Similarly, in SPFMV there is HCPro like in the rest of potyviruses, but the RSS function was associated to P1N-PISPO[9,10].

Purification of virions from infected susceptible plants often results in low yields, thus hampering structural studies. In some cases, the accumulation can be higher in experimental hosts compared to the natural hosts. Although SPMMV can infect several experimental hosts, in particular *N. benthamiana* and *N. tabacum* being quite adequate for virion purification, this is not the case for SPFMV, a virus with a rather narrow host range. This has hindered the direct comparison of the virions of the two viruses. Besides, SPFMV always shows very low accumulation in single infections of sweet potato plants, and the boost of accumulation in mixed infections with SPCSV prevents the preparation of pure virions, because both the potyvirus and the crinivirus partners have elongated flexuous particles that are likely to be very difficult to separate. To solve all these issues and allow structural studies, we have taken advantage of the similarities between virions and virus-like particles (VLPs) to use a recently described methodology for production of helical VLPs, based on expression of viral CPs from a heterologous replicating viral RNA that is also incorporated into the VLPs. This approach has been shown to be highly effective at producing VLPs that closely resemble the structure of the original virus[37]. Using this strategy, elongated flexuous VLPs of both SPFMV and SPMMV were efficiently produced *in planta*, allowing us to purify them in sufficient quantities for structural analysis to test the hypothesis that certain differences might serve to explain their biological peculiarities and vector specificity. Here we report the high-resolution CryoEM structures of SPFMV and SPMMV, two viruses that infect the same natural host, sweet potato, but belonging to two different genera *Potyvirus* and *Ipomovirus* within the family *Potyviridae*. This has allowed a comparison of their structural properties with the aim of gaining insights into their different modes of transmission.

## Results

**Production of VLPs *in planta* using the self-replicating pEff vector.** Constructs pEff-SPFMV-CP and pEff-SPMMV-CP, designed to express the CPs of SPFMV and SPMMV (Fig. 1a), were agroinfiltrated in *N. benthamiana* plants. In a previous work[37], a different CP sequence corresponding to another isolate of SPFMV (GeneBank: NC001841) was used to test the versatility of the pEff-based expression system for different VLPs. For that work the CP sequence was synthesized by a commercial provider, but in the present work we chose to construct a new version of pEff-SPFMV-CP amplifying the CP gene of a plant-infecting isolate (GeneBank: KU511268), in order to allow comparisons between VLPs and virions.

The effect of pEff-based constructs on the agroinfiltrated tissues was consistent with previous observations[37]. In our conditions, pEff-SPMMV-CP appeared to affect the tissue less severely than pEff-SPFMV-CP, which in some cases resulted in necrosis at late time points > 7 days after agroinfiltration (dpa).

The accumulation of CPs was determined by SDS-PAGE and western blot analysis in total protein extracts of agroinfiltrated plant tissue at 3, 5, and 7 dpa. Our results showed that both

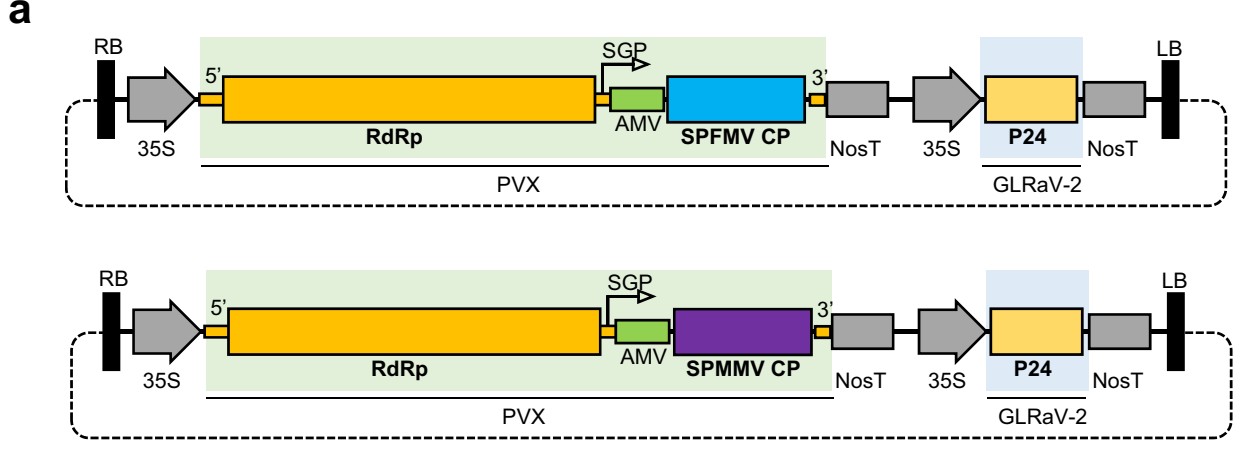

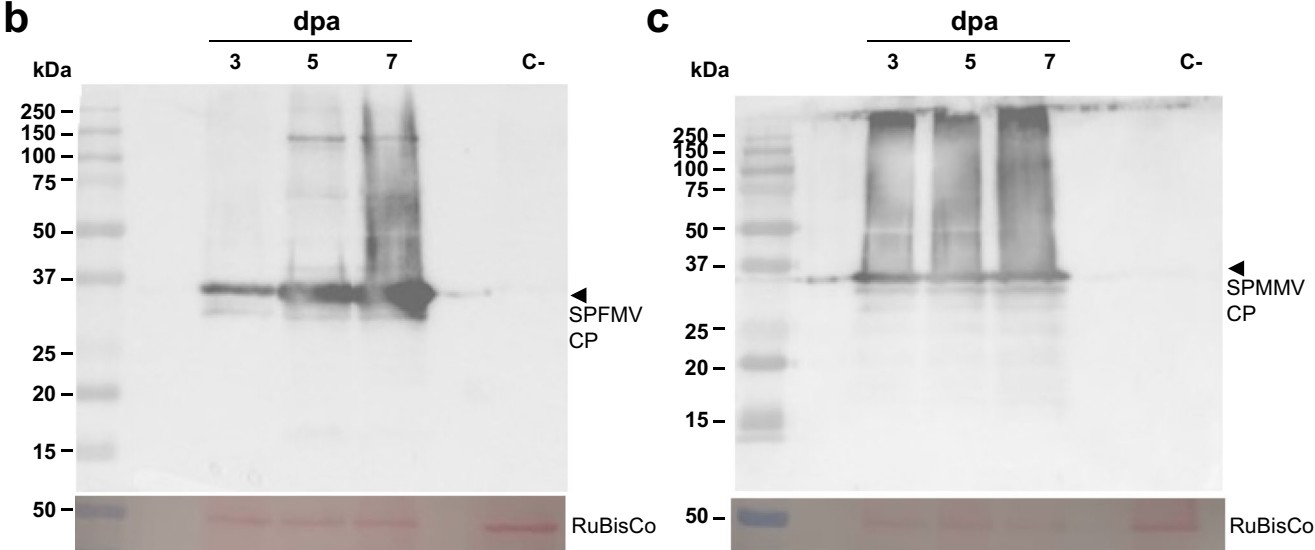

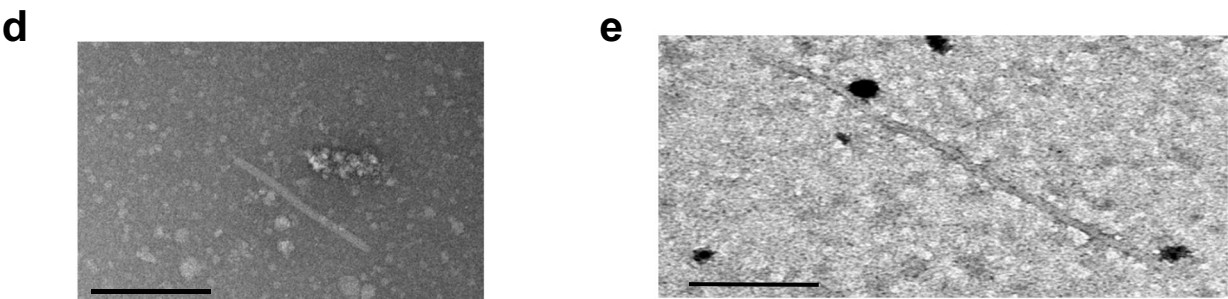

SPFMV and SPMMV CPs were detected as products with their expected sizes (Fig. 1b, c). This result was consistent in the case of SPFMV with those reported previously[37]. Comparatively, the amount of SPMMV-CP produced was similar for the three time points analysed, but SPFMV-CP appeared to accumulate more at late times after infiltration, roughly coincidental with the observed necrosis response. Unexpectedly, the polyclonal antibodies allowed cross-detection of the two CPs, despite only moderate amino acid identity between them (29% according to BLAST comparison), suggesting that some epitopes might be common for the two viruses in stretches of local similarity that can be identified in the alignment of the two sequences (Supplementary Fig. 1). TEM analysis of negatively stained samples revealed the presence of VLPs for both viruses (Fig. 1d, e) confirming that the pEff system provided an efficient platform for the production of potyvirid VLPs.

**Fig. 1 Transient expression of capsid proteins (CPs) and formation of Virus-like particles (VLPs). a** Schematic maps of pEff-based constructs with the subgenomic mRNA generated by the subgenomic promoter (SGP) in the cassette of the PVX replicon after the viral RNA-dependent RNA polymerase (RdRp). A separate cassette contains the RNA silencing suppressor P24 of GLRaV-2, both under the control of promoter (35S) and terminator (NOS) signals. Both parts are flanked by RB and LB Right and Left borders required for transfer of the T-DNA to the plant. **b** Western blot analysis of the expression of SPFMV CP (35 kDa, arrow) at 3, 5 and 7 days post agroinfiltration, using a specific polyclonal antibody. **c** Western blot analysis of the expression of SPMMV CP (34 kDa, arrow) at 3, 5 and 7 days post agroinfiltration, using a specific polyclonal antibody. In both **b** and **c**, samples infiltrated with the pEff empty vector were included as negative controls, and the panels below show the corresponding Ponceau S staining of the large subunit (53 kDa) of Rubisco protein as loading control. **d** Transmission electron microscopy (TEM) image of an assembled particle in clarified crude extract of plant tissue infiltrated with pEff-SPFMV-CP, captured using a Talos F200C TEM fitted with a Gatan OneView camera. Bar corresponds to 200 nm. **e** Transmission electron microscopy (TEM) image of an assembled particle in clarified crude extract of plant tissue infiltrated with pEff-SPMMV-CP, captured in a JEM-1400 microscope. Bar corresponds to 200 nm.

**Purification of VLPs**. For SPFMV, several previously published protocols for purification of SPFMV virions[38–40] proved inadequate or consistently resulted in low yields (calculated as mg per g of fresh weight tissue) when applied to VLPs. Despite extensive modifications aimed at improving the methods, yields remained around 0.04 mg/g. Nevertheless, we opted to use a modified procedure based on the protocol of Nakashima and coworkers[40], starting with more plant material to compensate for the low yield. After separation of the sample in sucrose density gradients and analysis by SDS-PAGE, several fractions showed a major protein of around 35 kDa as the expected size for CP (Fig. 2a). TEM analysis of these fractions showed the presence of flexuous VLP filaments with a diameter of 12–15 nm, but with lengths shorter than that expected according to Moyer & Kennedy[38] for native virions (around 850 nm). Our VLPs measured in the range of 60–700 nm (average 262 nm), with the most abundant sizes in categories around 200–250 nm and an additional secondary peak about 400 nm (Fig. 2b and Supplementary Fig. 2).

The purification of SPMMV VLPs was achieved with fewer difficulties following a published procedure for virion purification in phosphate buffer[28], reaching yields around 0.35 mg/g of fresh weight tissue. After fractionation in sucrose gradients, two products around 34 and 30 kDa were observed in the SDS-PAGE analysis. The two products were recognized by CP antibodies, suggesting that they might correspond to the full-length CP and a partially degraded product (Fig. 2c and Supplementary Fig. 3). In negative staining EM, VLPs were observed with a structure similar to the virions as flexuous filaments of 12–15 nm in diameter and lengths ranging from 60 to 550 nm (average 250 nm), with the most abundant sizes again in categories around 200–250 nm and 400–450 nm (Fig. 2d and Supplementary Fig. 2).

**Detection of RNAs present in VLPs**. Previous work showed that pEff-generated potexvirus and virgavirus VLPs contain RNAs derived from the replication of the vector, reinforcing the idea that an RNA scaffold was important for helical particle formation[37,41]. To assess whether this also applies for pEff-produced potyvirid VLPs, we analysed both types of purified SPFMV and SPMMV VLPs. RNA was extracted from VLP preparations, electrophoresed, transferred to membranes and hybridized with specific probes[42]. To confirm specificity, probes were also tested in heterologous combinations, finding that each probe hybridized only with RNAs present in the corresponding homologous VLPs, while no hybridization signals were observed in control lanes corresponding to RNA derived by tissue infiltrated with the empty pEff vector or a mock *N. benthamiana* plant, (Fig. 2e, f) where methylene blue-staining revealed RNA bands corresponding to the expected plant-derived ribosomal RNAs. The estimated size of the majority RNA components in samples of both SPFMV and SPMMV VLPs was above 1000 nucleotides, likely corresponding to the sub-genomic RNAs

dedicated to the expression of CP genes, confirming the expected high activity of the corresponding sub-genomic promoter. To evaluate whether other observed RNAs in VLPs were originated from the replicating vectors, a probe against the PVX replicase gene was also tested to specifically detect pEff-derived mRNA. This confirmed that the pEff-derived potyvirid VLPs contained RNA that originates from the replication of the pEff vector (Supplementary Fig. 4).

Regarding the sizes of encapsidated RNAs, the most abundant RNAs were expected to correspond to the subgenomic components expressing CPs, calculated to be >1100 nucleotides for both SPFMV-CP and SPMMV-CP, while the complete pEff-derived mRNAs were >5600 nucleotides. Considering the proportional sizes compared to virions, the expected sizes of VLPs would be around 100 nm for the sgRNAs and 470 nm for the full-size replicating mRNAs. These two sizes were compatible with our measurements of VLPs, although the distribution did not fit exactly a bimodal curve (see again Supplementary Fig. 2).

**Thermal stability of VLPs**. Differential scanning fluorimetry was employed to assess the thermal stability of the VLPs, using Sypro Orange as an extrinsic fluorophore that interacts with exposed hydrophobic residues[43,44]. Two shifting temperatures were determined for each type of VLPs (Fig. 3a, b), consistent with two transitions, likely corresponding the first to the dissociation of the VLP into its constituent CP monomers ($T_{m1}$), and the second to the unfolding of the CP subunits ($T_{m2}$). The fluorescence signal obtained for SPFMV-VLPs was significantly lower than for SPMMV-VLPs, indicating different interactions between the fluorophore and the VLP/CP for each system. The transition temperatures were calculated after data normalization using the first derivative of a native fraction for each sample (Fig. 3c), and the results showed that SPFMV-VLPs dissociated at 55 °C ($T_{m1}$), while the CP units unfolded at 66 °C ($T_{m2}$), whilst SPMMV-VLPs dissociated at 60 °C ($T_{m1}$), with unfolding of CP units at 69 °C ($T_{m2}$). Altogether these values further proved that the ipomovirus VLPs exhibited a measurably higher stability compared to the potyvirus VLPs.

**Architecture of SPFMV and SPMMV**. The structures of SPFMV and SPMMV were determined using single particle cryoEM with helical symmetry, at resolutions of 2.6 Å (SPFMV) and 3.0 Å (SPMMV) (Fig. 4 and Table 1). The overall architecture of both SPFMV and SPMMV VLPs was similar to that determined for other plant flexuous filamentous viruses[21–23]. Both structures form a left-handed helical arrangement, made up of around 8.8 subunits per turn with a diameter of 130 Å. Each subunit is separated by helical rises of 3.97 Å and 3.98 Å and helical twists of 41.2° and 40.83°, respectively, for SPFMV and SPMMV. Comparing the two structures overall, the helical organisation is

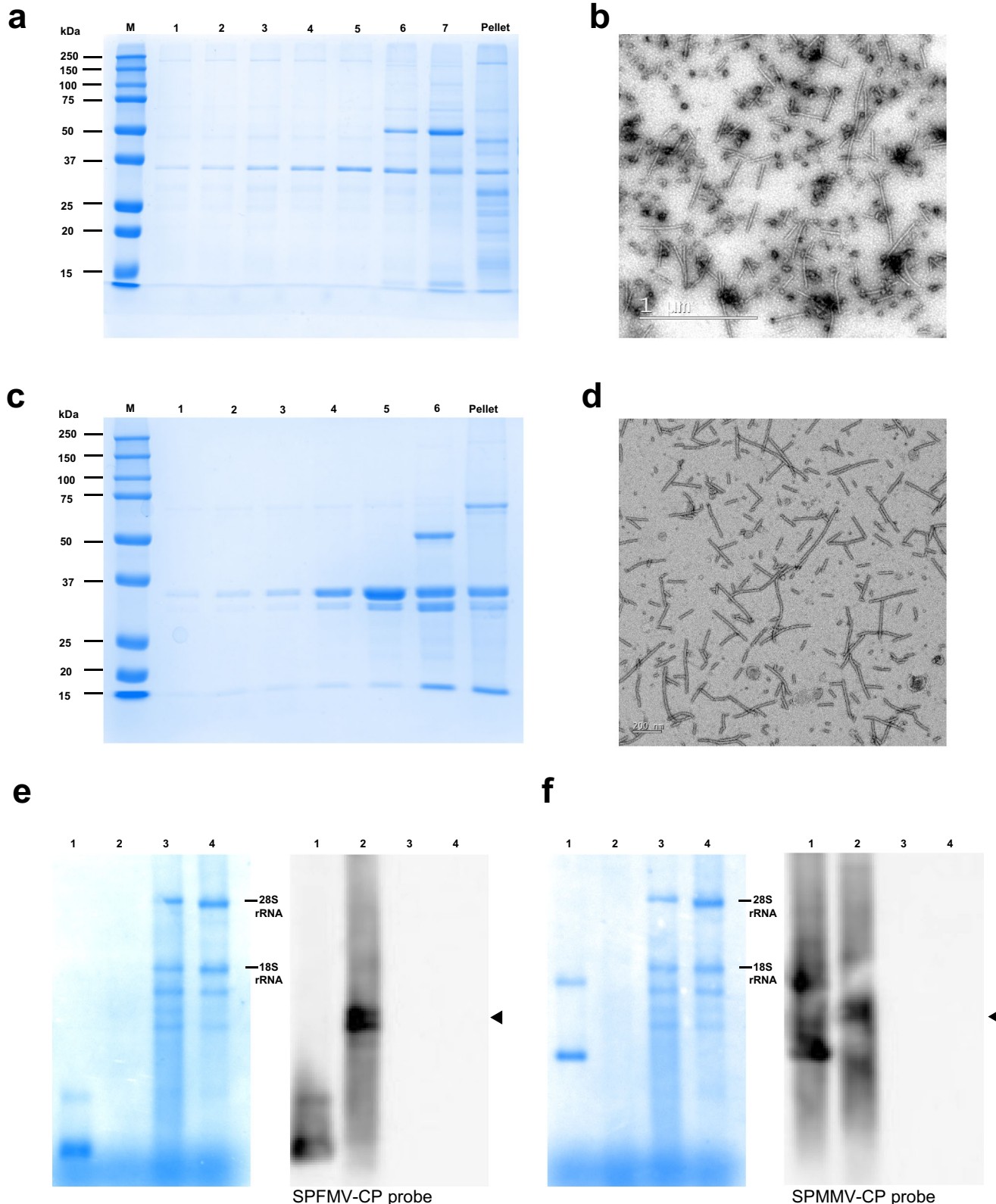

relatively similar. Small variations within the three domains of the individual coat proteins are apparent.

The cryoEM maps of the sweet potato viruses allowed an atomic model to be built for most of the coat protein sequence. Owing to the flexible nature of the N- and C-termini, parts of these segments lacked sufficient resolution to model the precise sequence but were modelled as polyalanine wherever the

polypeptide backbone could be traced. Coat proteins from both structures contain three domains; an N-terminal domain made up of 1–123 (SPFMV), 1–111 (SPMMV) residues, a core domain with residues 124–276 (SPFMV), 112–262 (SPMMV) and a C-terminal tail comprising 277–316 (SPFMV), 263–305 (SPMMV) residues. The cryoEM density allowed residues 91–310 (SPFMV) and residues 66–296 (SPMMV) to be modelled,

**Fig. 2 Purification of SPFMV and SPMMV VLPs, and detection of protein and RNA components. a** SDS-PAGE of samples corresponding to sucrose gradient fractions (40–10%, lanes 1–7) of purified SPFMV-VLPs. A single band of ~35 kDa is detected with the expected size of SPFMV-CP, and only in fractions 6 and 7 with an additional band (50 kDa) probably being the large subunit of Rubisco. **b** TEM analysis of fraction 5. Scale bar corresponds to 1 µm. **c** SDS-PAGE of samples corresponding to sucrose gradient fractions (40–10%, lanes 1–6) of purified SPMMV-VLPs, showing a prominent band that coincides with SPMMV-CP size (34 kDa, while the shorter band around 30 kDa might probably be a truncated form). In fraction 6, there is an additional band (50 kDa, likely being the large subunit of Rubisco). **d** TEM analysis of fraction 4. Scale bar corresponds to 200 nm. **e** Methylene blue staining (left) and Northern blot analysis (right) of RNA samples extracted from purified SPFMV-VLPs (lane 2), *N. benthamiana* plant tissue agroinfiltrated with the corresponding pEff vector (lane 3), and tissue from a mock plant (lane 4). The Northern blot was incubated with a SPFMV-CP specific probe. A PCR product corresponding to the region used for the probe was included as control for the hybridization (lane 1), and the mobilities of 28S and 18S rRNAs, corresponding respectively to 4000 and 1600 nts, are indicated as internal size markers. The arrowhead indicates the major RNA found in the VLPs. **f** Methylene blue staining (left) and Northern blot analysis (right) of RNA samples extracted from purified SPMMV-VLPs (lane 2), plant tissue agroinfiltrated with the corresponding pEff vector (lane 3), and tissue from a mock plant (lane 4). The Northern blot was incubated with a SPMMV-CP specific probe. A PCR product corresponding to the region used for the probe was included as control for the hybridization (lane 1), and the mobilities of 28S and 18S rRNAs, corresponding respectively to 4000 and 1600 nts, are indicated as internal size markers. The arrowhead indicates the major RNA found in the VLPs.

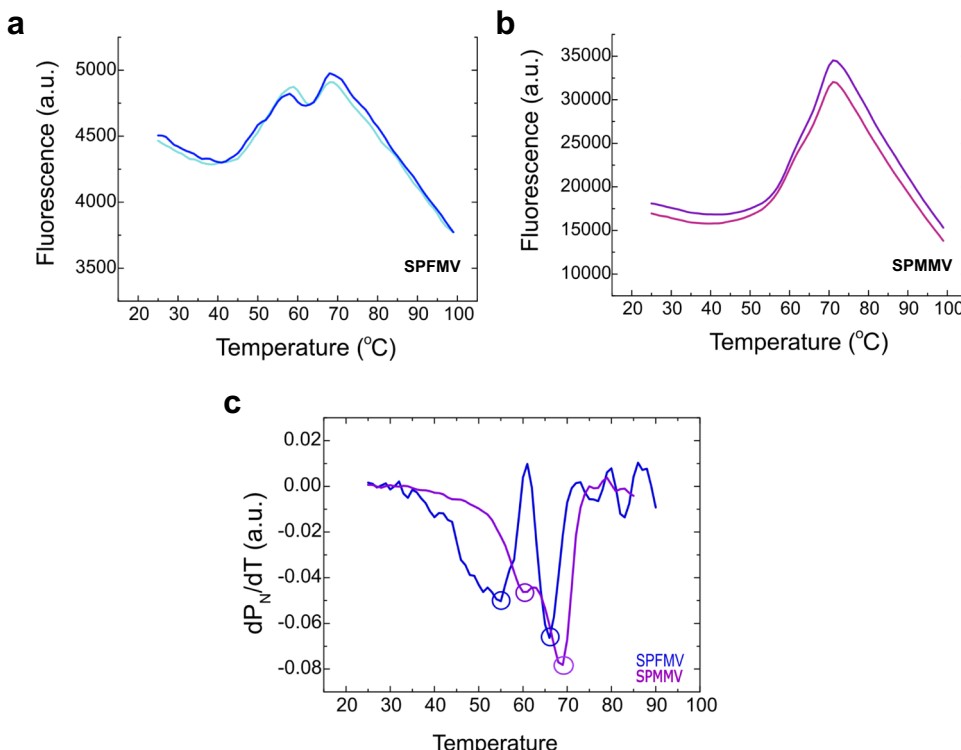

**Fig. 3 Thermal stability assay of purified VLPs of SPFMV and SPMMV. a** Differential scanning fluorimetry analysis revealing 2 shifting temperatures for SPFMV-VLPs. **b** Differential scanning fluorimetry analysis revealing 2 shifting temperatures for SPMMV-VLPs. **c** First derivative graphical representation of the melting temperatures for the two types of VLPs, after data normalization. For each type of VLPs, the first shifting temperature (Tm1) accounts for the particle's oligomeric dissociation, while the second (Tm2) represents most probably the CPs unfolding.

identifying the major components of the three domains in each CP structure. The core domain consists of nine α-helices (SPFMV) and two β-strands, a feature which appears to be conserved amongst other viruses in the *Potyviridae* family. An RNA binding pocket is situated at a crevice formed by the core domain and the C-terminal tail. The C-terminus in the filament forms a spiral arrangement, resulting in the formation of an inner tube of density, formed by the extreme C-terminus of the CPs.

**Inter-subunit interactions**. In both structures, the subunits are arranged to form a helical screw shape that forms the capsid. The N-terminal portion from each CP connects to the adjacent subunit around the helical screw, as well as to subunits in the next layer 'down' (Fig. 5). Similar to the other *Potyviridae* structures, the N-terminus takes a 90° turn after connecting to the adjacent

subunit in order to hold the subunit facing downwards by a series of interactions. This arrangement is strikingly different to *Alphaflexiviridae* structures where each CP N-terminus only connects the adjacent subunit[45]. A series of hydrogen bonds and salt bridges in both structures hold the subunits together tightly (Fig. 5; Supplementary Tables 1, 2).

The architecture of the N-terminal arm as well as the mode of interaction is conserved between the two structures, with local variations observed based on the sequences. For instance, α-helix 1, which slots in the crevice of the adjacent subunit is tilted downwards in SPMMV, compared with SPFMV and PVY N-terminal regions (Supplementary Fig. 5), with a root-mean square deviation (RMSD) of 4.7 Å. Overall, the high degree of similarity between the coat proteins of the two viruses suggests a common mode of interaction between CPs during capsid assembly.

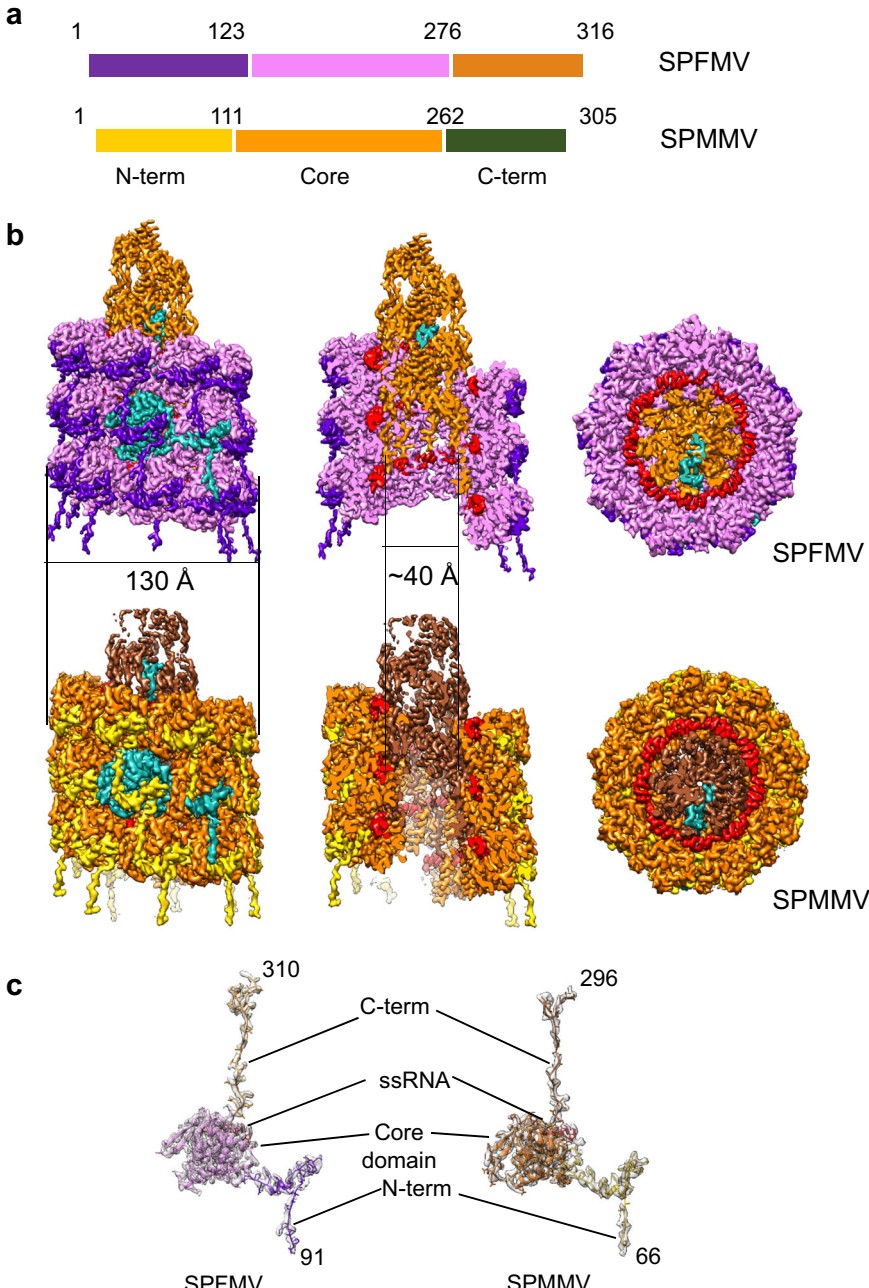

**Fig. 4 CryoEM structures of *Potyviridae*-VLPs. a** Schematic organization of SPFMV and SPMMV VLPs, colour coded according to the three domains and number of residues indicated for the corresponding coat protein sequences. **b** CryoEM maps for the indicated VLPs of SPFMV and SPMMV. The maps are coloured according to the three domains in the single coat protein used in (**a**) with one of the coat proteins in both structures coloured in cyan. Both panels show the cryoEM maps in side-view (left), cut-away view (middle) and top view (right). The ssRNA is coloured in red, and the outer and inner diameters are labelled. **c** CryoEM map fitted structural models for individual coat proteins from SPFMV (left) and SPMMV (right) are shown, labelled with coat protein regions.

In order to model the extreme N- and C-termini regions, not sufficiently well resolved in the cryoEM structures due to the weak signal that results from averaging different conformations, a sequence-based structure prediction for the two viruses was done using AlphaFold (Supplementary Fig. 6). The best scoring predicted models recapitulated the cryoEM structures of the CP with high degree of similarity in the core domains. Interestingly, the predicted structure for SPFMV CP showed a largely unstructured N-terminus, supported by the high b factor in this region and suggested reduced accuracy in prediction in this region. On the other hand, the predicted structure for SPMMV

CP showed a flanking loop followed by helix-turn helix motif at the N-terminus, not observed in the cryoEM structure. Considering the high confidence score in this region of AlphaFold predicted structure, it is likely that this structural motif is averaged out during cryoEM 3D reconstruction due to the possible high flexibility of the N-terminal loop preceding it.

**Interactions between the coat protein subunits and incorporated RNAs.** In the high-resolution CryoEM structures for SPFMV and SPFMV, five nucleotides of the encapsidated ssRNA

**Table 1 CryoEM data analysis report.**

| | SPFMV | SPMMV |
|---|---|---|
| **Data collection and processing** | | |
| Magnification | 75,000× | 75,000× |
| Voltage (kV) | 300 | 300 |
| Total electron exposure (e-/Å²) | 40.88 | 42.47 |
| Detector (mode) | Falcon III (counting) | Falcon III (counting) |
| Defocus range (μm) | −0.5 to −3.0 | −0.5 to −3.0 |
| Pixel size (Å/pixel) | 1.065 | 1.065 |
| Initial no. of micrographs | 3276 | 2260 |
| Final no. of micrographs | 3276 | 2260 |
| Initial no. of particle images | 313.330 | 329.573 |
| Final no. of particle images | 48.042 | 30.663 |
| Map resolution (Å), 0.143 FSC | 2.6 | 3.0 |
| Helical Symmetry imposed (rise, twist) | C1: −3.98, 40.9 | C1: −3.97, 41.29 |
| Map sharpening B factor (Å2) | −140.9 | −123.7 |
| **Refinement** | | |
| Initial model used | | |
| Model resolution Å | 2.7 | 3.1 |
| Model composition (1 CP) | | |
| Non-hydrogen atoms | 1813 | 1784 |
| Protein residues | 220 | 231 |
| Nucleotides | 5 | 5 |
| Ligands | 0 | 0 |
| Validation | | |
| MolProbity score | 1.73 | 2.11 |
| Clashscore | 4.83 | 7.76 |
| Poor rotamers (%) | 0 | 1,32 |
| Ramachandran plot | | |
| Favoured % | 92.2 | 88.21 |
| Allowed % | 7.8 | 11.79 |
| Disallowed % | 0 | 0 |

were observed in the binding pockets of individual coat proteins. The resolution of the maps was sufficient to build the five nucleotides RNA chain de novo with uridine nucleotides. The exact RNA sequence is difficult to discern as the densities for the RNA observed are a result of the averaged segments from the different parts of the virus filament. Both structures show the RNA spanning a length of 23 Å, corresponding to a major part of the inner core domain of the coat proteins as well as a similar conformation with an RMSD of 0.7 Å.

The RNA segments are bound to an inner RNA binding pocket, formed by residues from the core domain and the C-terminal segment of the coat protein (Fig. 6). This is a largely charged pocket formed by arginines and lysines that bind and interact with the negatively charged RNA backbone, as well as making a series of hydrogen bonds (Fig. 6; Supplementary Table 3). Similar to other *Potyviridae* structures, three residues (Arg, Ser, Asp) are found to be conserved at this binding pocket which are believed to be important in correct packaging of the RNA during viral assembly as shown also in the case of the potexvirus PepMV[21,46]. In the sweet potato virus structures, these correspond to Arg 235, Ser 172 and Asp 248 (SPFMV), and Arg 262, Ser 152 and Asp 236 (SPMMV) respectively (Fig. 6). Another conserved feature observed in the two sweet potato virus structures is the fourth nucleotide of the RNA having its nucleotide base facing inwards towards the pocket. This appears to be a common feature observed in other *Potyviridae* and

*Alphaflexiviridae* structures, suggesting the importance of this interaction within the RNA binding pocket amongst the plant viruses (Supplementary Table 3). Together, the cumulative effect of the interactions at the RNA binding pocket from the core domain of the CP shields the RNA from damage.

## Discussion

The economic impact of crop diseases caused by potyvirids requires research to develop efficient control strategies. Among others, the molecular interactions between viruses and insect vectors are highly relevant for the purpose of reducing virus dispersal[47]. Here we determined the structure of virions of two different *Potyviridae* members using RNA-containing VLPs produced by a replicating PVX-based viral vector[37]. This technology allowed us to explore the structures of the potyvirus SPFMV and the ipomovirus SPMMV, infecting the same host but belonging to genera transmitted by different insect vectors.

Derived atomic models confirmed the assembly of helical filaments resembling previously published structures of potyvirus virions[21–23]. Consistent with previous publications, both types of VLPs contained vector-derived RNA scaffolds for particle formation[37,48]. Under certain conditions, previously published investigations showed that the CPs of some potyvirids were able to self-assemble apparently in absence of RNA[49,50], and indeed different RNA-devoid potyvirus and potexvirus VLPs have been produced[22,51–53]. Nonetheless, in some cases, such as PVY, the absence of RNA compromised the precise helical symmetry, resulting instead in stacked-ring filaments, revealing again the crucial role of RNA in ensuring proper virion assembly[22]. Similarly, TuMV VLPs lacking RNA were less stable and highly heterogeneous compared to wildtype virions, with poor structural details that precluded the construction of an accurate atomic model[23]. Moreover, compared to the reported yield of empty TuMV VLPs (around 10 μg/g)[52], our pEff-mediated system generated higher yields of VLPs, ranging from 4 to >30 times respectively for SPFMV and SPMMV, suggesting that production of VLPs was indeed facilitated. Observations on the distribution of VLP lengths do not fit exactly with the expected sizes of the most abundant RNAs derived from the replicative pEff-vector constructs. In other cases the predicted equivalences were more evident, as found with the tobamovirus TMV[41] and with the potexvirus AltMV[37], suggesting that for the sweet potato potyvirids the encapsidation of RNAs into VLPs was less strictly regulated. Alternatively, we cannot exclude the possibility that the conditions required for purification might result in partial breakdown of VLPs due to shearing forces.

Although belonging to the same family and sharing a similar structural organization, SPFMV and SPMMV particles presented a clear divergence in terms of stability, as reflected by their properties during the purification processes. Whereas SPMMV VLPs were readily purified and maintained integrity during long-term storage at low temperatures (4 °C), this was not the case for SPFMV VLPs, where particles were apparently easily disassembled and/or degraded, likely contributing to the much poorer purification yields. However, it is possible that differences in aggregation properties might result in losses during early stages of purification. When submitted to differential scanning fluorimetry to analyze thermal stability, the two types of VLPs exhibited measurable differences, with SPFMV VLPs being less stable than SPMMV since they were disassociated at lower temperatures. Furthermore, differences were also observed in the second thermal shifts, probably corresponding to denaturation of monomeric CPs. Recent investigations are revealing the importance of structural characteristics, such as stability and dynamics of virions, in relation to biologically relevant functions of different viruses[54].

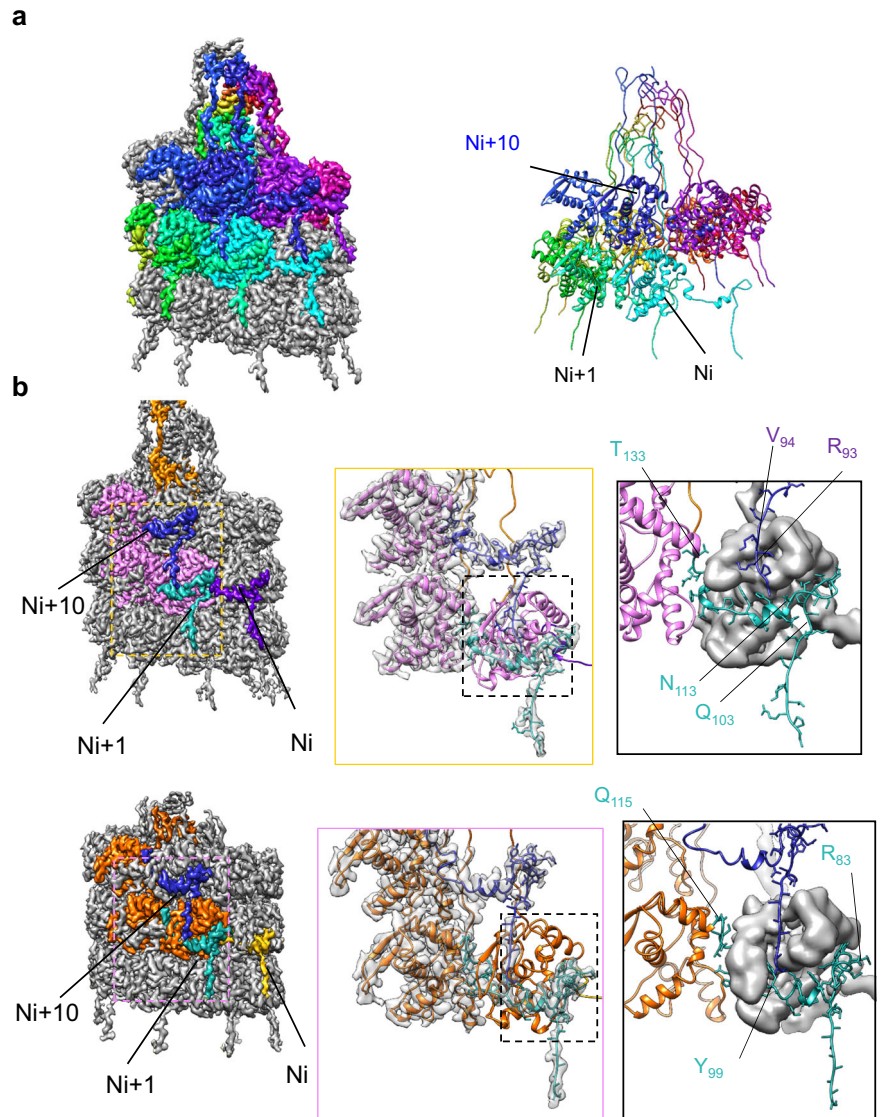

**Fig. 5 Subunit arrangement in the sweet potato VLPs. a** CryoEM map of SPFMV coloured according to the subunit elevation along a helical turn (starting subunit in cyan and 10th subunit in dark blue). Right panel shows the corresponding structural model of one complete segment with first, second and 10th subunits labelled. **b** CryoEM map of SPFMV (top panel) coloured according to N-termini of first CP (in purple), N + 1 (in cyan) and N + 10 CP (in dark blue) along the helical segment. The three CPs are coloured in pink with the C-terminus in orange. Bottom panel shows cryoEM map of SPMMV, coloured according to N-termini of first CP (in gold), N + 1 (in cyan) and N + 10 (in dark blue). Middle panels show a close-up view of the N-terminus organization between Nth, N + 1 and N + 10 CP in both SPFMV and SPMMV structures. Right panels show the outlined, zoomed-in area for N-terminal interaction with Nth CP via N + 1 (cyan) and N + 10 (dark blue) N-terminus in SPFMV (top right) and SPMMV (bottom right) respectively. Key residues are labelled with the Nth CP shown as gaussian filtered map to illustrate binding pockets, N + 1 model in pink (SPFMV) and in orange (SPMMV), with their N-terminal segments in cyan and the N + 10 N-terminal segments shown in dark blue.

Regarding the integrity of CPs, it is known that endogenous plant peptidases are responsible for the susceptibility of the N-terminal region of many potyviral CPs to proteolytic cleavage[55]. We confirmed the occasional presence of CP-related products with faster mobilities in SDS-PAGE than the expected for intact CPs (Supplementary Fig. 4). To minimize the damage, a Protease Inhibitor Cocktail was incorporated to our purification protocols. However, despite reducing the CP proteolytic degradation, this treatment did not modify the yields of VLPs, which for SPFMV always remained below those of SPMMV, probably reflecting other structural differences. Again, intrinsic solubility with less aggregation might help to explain the consistently higher yields obtained for SPMMV VLPs.

Our data suggest that SPFMV particles are more fragile than those of SPMMV. This could influence their vector specificity

(aphids and whiteflies, respectively)[18,56] though further studies would be needed to confirm this.

Computational analysis has classified potyviral CP as one of the most intrinsically disordered proteins among the virus-encoded gene products[57], a feature probably linked to its functional versatility, enabling multiple interactions with other virus, host or vector factors[13]. Our cryoEM data revealed a high resemblance between SPFMV and SPMMV atomic structures, adopting a similar architecture compared with other poty- and potexviruses[21–23,46,58]; however, in virtually all available structures, the first N-terminal residues could not be traced (in this study, 90 aa for SPFMV and 65 aa for SPMMV). The N-terminus comprises the most variable region within potyviral CPs, where there is a conserved DAG motif related to aphid-mediated virus transmission, with the assistance of HCPro[14,16]. On the other hand, no specific motifs nor vector

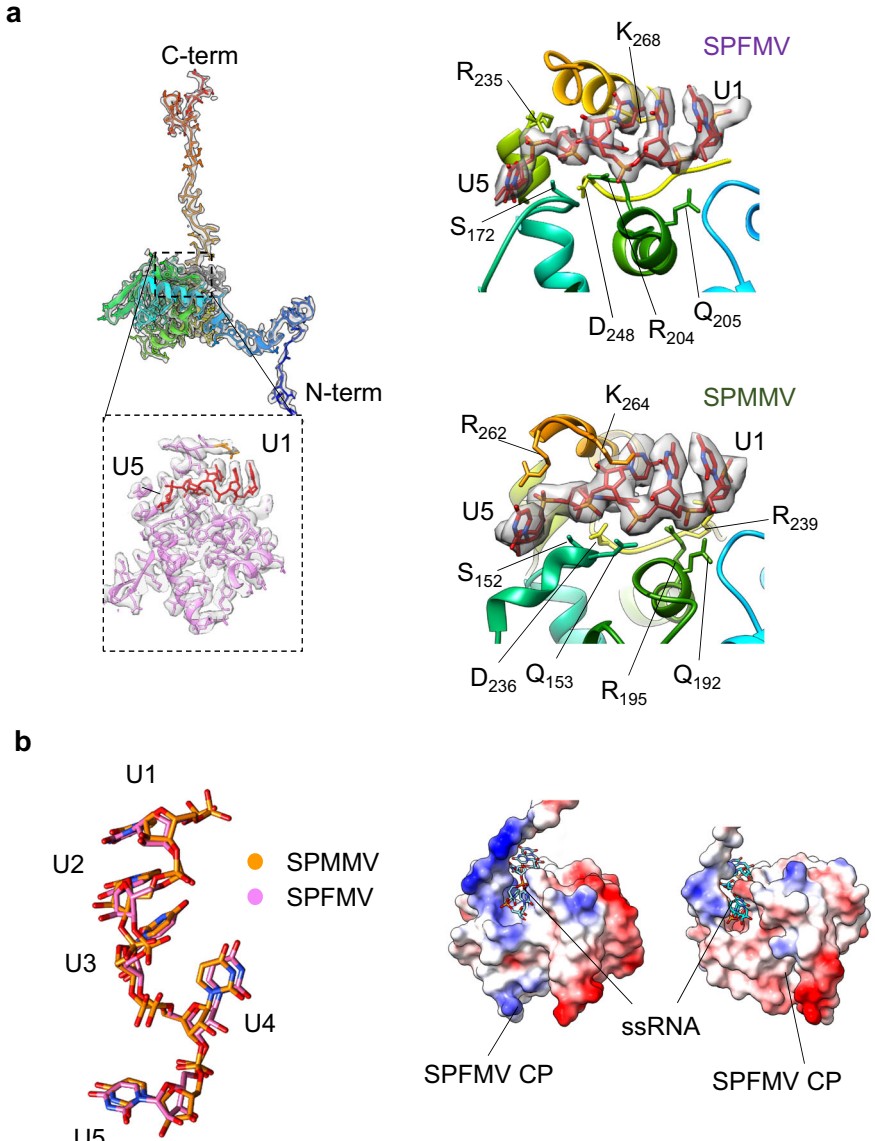

**Fig. 6 ssRNA interactions. a** Left panel shows fitted atomic model of SPFMV CP, coloured from N- (blue) to C-termini (red) with cryoEM densities shown in grey. A zoomed-in panel shows the RNA binding cavity located within the core domain; SPFMV model shown in pink with RNA nucleotides U1 to U5 in red. Top right and bottom right panels show the RNA binding site in SPFMV and SPMMV respectively, with fitted atomic coordinates for ssRNA (in red) inside cryoEM map (shown in grey). Key interacting residues are labelled. **b** Left panel shows the aligned structures of ssRNA from SPFMV (in pink) and SPMMV (in gold) with labelled nucleotides. Right panel shows the computed surface electrostatic potential maps for CPs, viewed along the RNA binding pockets, with the ssRNA shown as an atomic model in cyan. The colouring ranges between positively charged (blue), neutral (white) and negatively charged (red) spots.

receptors have been identified to date accounting for *B. tabaci* transmission of ipomoviruses, although recent investigations on CP mutational analysis of a CVYV clone showed a clear implication of the protein N-terminus in whitefly transmissibility[59]. Considering the difficulties in experimentally transmiting SPMMV by *B. tabaci*[28,60] it would be particularly interesting to further explore the implication of CP during this process. It is tempting to consider that the high-confidence prediction by AlphaFold of a structured section near the N-terminus of SPMMV CP might be relevant for transmissibility and/or other viral functions. In this line, predictions of CPs of other ipomoviruses also resulted in the identification of distinctive structural elements near the N-terminus, a conserved feature that might play a role in the transmission process: further experimentation will be needed to confirm that hypothesis.

In contrast to the highly variable N-terminal region, the central core region of the coat protein is well conserved[61], showing a common pattern among many other filamentous viruses, some even belonging to distantly related families infecting animals[46,62]. The complete structural resolution of the CP core for both viruses, allowed us to determine common features with other potyviruses, including the position of three key residues (Arg, Ser and Asp) apparently responsible for the RNA binding in the predicted pocket spanning five RNA nucleotides in our structures (Fig. 6) and conserved in other potyviral CPs[21–23]. Notably, these three amino acids are present in equivalent positions in both CPs of the potyvirus and the ipomovirus, suggesting that the RNA binding pocket is not limited to potyviruses but it is a shared feature among potyvirids. These conserved residues might play key roles also during the assembly process of VLPs[63,64]. In

summary, the interaction of RNA and CP in potyvirids seems to be quite conserved, becoming an attractive target for antiviral drugs that could disrupt virion formation.

The density maps of both sweet potato VLPs allowed mapping of nearly the complete CP C-terminus, only missing the last 6 and 9 aa for SPFMV and SPMMV, respectively. In both particles the C-terminus appeared inside the viral lumen, an observation contrasting with earlier predictive models based on immunogenicity and proteolytic treatments with trypsin that proposed that both N- and C-terminal regions of the potyviral CPs were surface exposed[65]. However, all recent structural data have showed that the folding of potyviral CPs results in the C-terminal region being located at the inner surface as a tube where molecules of certain sizes could have access[22]. Our results fully agree with these data, with an internal diameter of about 40 Å. Topologically speaking, if the inner tube is accessible to proteases, then the C-terminal region can be indeed exposed, although the difference with the earlier models[65] will be the external/internal placements of N- and C-terminus. The involvement of the CP distal arms (N- and C-) in the virion structure have been addressed for empty TuMV VLPs, demonstrating that deletions affecting either the N-terminal protrusions or the C-terminal distal regions do not interfere with CP accumulation or VLPs assembly. This suggests that these parts might be dispensable during particle formation[66]. The same study proposed a direct role of the C-terminal domain in particle length determination; however, this did not apply to PVY VLPs, reflecting possible differences between potyvirus species in that concrete region[22].

Overall, in both SPFMV and SPMMV structures, CP polymerization appears facilitated by side-to-side and axial connections of the N-terminal part of each CP subunit, a feature that seems universal to *Potyviridae* members[21–23] but differs for *Potexvirus* members, where only adjacent CPs seem to interact[45,46], likely suggesting a less compact and stable structure for *Alphaflexiviridae* species.

Our study contributes novel structures of two sweet potato potyvirids from the aphid-transmitted genus *Potyvirus*, and the whitefly-transmitted genus *Ipomovirus*. The structures allow a direct comparison of the coat protein organization and the bound RNA, structural features that are key to virus infectivity and mobility. Besides providing a basis for further investigations of viral pathogens in this important crop, our work also allows comparisons of virions transmitted by two different insect vectors. This could assist in the development of better tailored control measures against sweet potato viral diseases with different vector specificity. Moreover, the production of flexuous VLPs might be potentially useful in nanobiotechnology[67,68].

## Methods

**Experimental design**. The objective of our work was to produce VLPs of two viruses infecting sweet potato to assist in solving their structures, cloning the corresponding gene products in the pEff-based expression systems. Virus isolates of SPFMV[9] and SPMMV[36] were available in our laboratory. SPFMV-infected *I. batatas* and SPMMV-infected *N. tabacum* leaves were used to extract total RNA with TRIzol reagent (Invitrogen™) according to the manufacturer's instructions. Amplification of SPFMV-CP (GeneBank: KU511268) and SPMMV-CP (GeneBank: GQ353374) was done with Phusion High-Fidelity polymerase (Thermo Scientific™) after reverse transcription with MultiScribe™ (Invitrogen™, Thermo Fisher Scientific) and specific primers (listed in Supplementary Table 4). During amplifications, start codons (ATG) and AscI restriction sites were incorporated upstream of the CP genes, and also stop codons (TGA) and XmaI restriction sites were added downstream. This strategy[69] allowed cloning of CPs to the pEff-GFP expression vector by restriction enzyme digestion and T4 DNA ligase (Thermo Scientific™). The derived pEff-SPFMV-CP and pEff-SPMMV-CP plasmids were selected after heat-shock transformation of *Escherichia coli* (Top10 strain) and purification with the GeneJET Plasmid Miniprep Kit (Thermo Scientific™). Plasmid constructs were confirmed by Sanger capillary sequencing.

**Transient expression, protein extraction and western-blot analysis**. Once transformed into *A. tumefaciens* (LBA4404 strain) the plasmid constructs were infiltrated to *N. benthamiana* following previously described techniques[70–72]. Samples of infiltrated leaves were taken at 3, 5 and 7-days post agroinfiltration, and the expression of CPs was confirmed by Western blot analysis after grinding 2 leaf discs (~60 mm diameter) of infiltrated tissue in a TissueLyser II disruptor (QIAGEN, Hilden, Germany), homogenizing in 200 µl extraction buffer (20 mM Tris-HCl, pH 7.5, 30 mM NaCl, 1 mM Ethylenediaminetetraacetic acid, 0.5% (v/v) NP-40, 2% (v/v) β-mercaptoethanol). Clarified aliquots of supernatant after centrifugation (8000 × $g$) for 10 min at 4 °C were heated for 5 min at 95 °C with Laemmli loading buffer 4× (250 mM Tris-HCl, pH 7.5, 40% (v/v) glycerol, 8% (w/v) SDS, and 20% (v/v) β-mercaptoethanol) and electrophoresed in a 12% polyacrylamide gel with 1× running buffer (10× Tris-Glycine buffer: Glycine:1.92 M; Tris: 0.25 M; SDS 1%). InstantBlue (Abcam, Cambridge, UK) was used for gel staining and total protein quantification. For western blot detection of CPs, proteins were transferred to a nitrocellulose membrane (Amersham, GE Healthcare, Chalfont St Giles, UK), incubated first with polyclonal anti-SPFMV-CP or anti-SPMMV-CP (RT-0898 and RT-0900 respectively, DMSZ, Braunschweig, Germany) followed by Goat anti-rabbit horseradish peroxide (HRP) secondary antibody (Cat. 31460, Invitrogen™, Thermo Fischer Scientific), and finally revealed with chemiluminescence substrate (SuperSignal™ West Femto, Thermo Fischer Scientific) in a ChemiDoc™ imaging system (BioRad Laboratories, Hercules, USA).

**Detection of VLP assembly in planta**. Infiltrated tissue was either used fresh or frozen in liquid nitrogen. For pEff-SPFMV-CP the homogenization was done in 100 mM Sodium Borate pH 7.95 followed by low-speed centrifugation (8000 × $g$) for 15 minutes at 4 °C, and the supernatant was analysed after negative staining in Transmission Electron Microscopy (TEM) to visualize assembled particles (see details below). Similarly, for pEff-SPMMV-CP the infiltrated tissue was homogenized in phosphate-buffered saline (PBS) pH 7.2 and particles were detected in the derived supernatant after low-speed centrifugation (12,000 × $g$) for 15 min at 4 °C.

**Purification of SPFMV-VLPs**. SPFMV-VLPs purification was based on the protocol established by Nakashima and colleagues for virion purification[40], with several modifications. *N. benthamiana* infiltrated tissue was harvested at 7 days post agroinfiltration, frozen and homogenized in a laboratory waring blender (Waring Lab, Torrington, USA) with 3 volumes of extraction buffer #1 (50 mM Hepes buffer, 500 mM Urea, 50 mM EDTA, 0.5% Na$_2$SO$_3$, pH 8), containing Mini Complete™ EDTA-free Protease Inhibitor Cocktail (Merck KGaA, Darmstadt, Germany), trying to minimize degradation by protease activity. The homogenate was filtered through 2 layers of Miracloth (Merck KGaA, Darmstadt, Germany) and clarified at low-speed centrifugation (8000 × $g$) for 10 min at 4 °C, and the supernatant was stirred for 1 h with 2% Triton X-100 (Merck KGaA, Darmstadt, Germany) at 4 °C. After a second clarification through low-speed centrifugation (8000 × $g$, 10 min, 4 °C) the supernatant was deposited on the top of a 20% (w/v) sucrose cushion in extraction buffer #2 (20 mM Hepes buffer, 250 mM Urea, 10 mM EDTA, pH 7.2) and centrifuged at 130,000 × $g$ for 2.5 h at 4 °C in a swinging bucket SureSpin 630 Rotor (Thermo Scientific, Waltham, USA). Pellets were resuspended overnight in extraction buffer #3 (10 mM Hepes buffer, 50 mM Urea, 1 mM EDTA, pH 7.2). After clarification at 2000 × $g$ for 10 minutes at 4 °C the supernatant was loaded on a sucrose gradient 10%-40% w/v in extraction buffer #3 and subjected to ultracentrifugation at 90,000 × $g$ for 2 h at 4 °C, in a swinging bucket AH-650 Rotor (Thermo Scientific, Waltham, USA). Fractions of 500 µl were collected and their contents were analysed in a 12% SDS-PAGE gel, stained with InstantBlue (Abcam, Cambridge, UK) and also observed under TEM imaging. For cryoEM studies, fractions with higher concentration of VLPs were pooled and the excess of salt removed using PD-10 desalting columns (GE Healthcare, Chalfont St Giles, UK) equilibrated with extraction buffer #3 and further concentrated by spinning at 103,000 × $g$ for 1 h at 4 °C in AH-650 Rotor. The pellet was resuspended in extraction buffer #3, clarified at 8000 × $g$ for 5 min and the resulted supernatant was used for all subsequent analysis including cryoEM.

**Purification of SPMMV-VLPs**. SPMMV virus-like particles purification was based on previously described protocols[28] with modifications. Frozen *N. benthamiana* infiltrated tissue harvested at 5 days post agroinfiltration was homogenized in 3 volumes of 50 mM Sodium Phosphate buffer (pH 7, containing 0.1% β-mercaptoethanol) and clarified at low-speed (8000 × $g$, 15 min, 4 °C) before been subjected to ultracentrifugation (100,000 × $g$, 90 min, 4 °C) in a swinging bucket SureSpin 630 Rotor (Thermo Scientific, Waltham, USA). The pellet was resuspended overnight in phosphate buffer, clarified (2000 × $g$, 10 min, 4 °C) and loaded on a sucrose gradient 10–40% w/v for ultracentrifugation (100,000 × $g$, 3 h, 4 °C), in a swinging bucket AH-650 Rotor (Thermo Scientific, Waltham, USA). Fractions of 500 µl were collected and analysed in a 12% SDS-PAGE gel, stained with InstantBlue (Abcam, Cambridge, UK) and subsequent TEM imaging. For cryoEM studies, fractions with the higher VLPs-content were pooled and the excess of salt removed using PD-10 desalting column (GE Healthcare, Chalfont St Giles, UK) equilibrated with sodium phosphate buffer. As above, samples were submitted to cryoEM and other analysis.

**RNA extraction from VLPs and northern blot analysis**. RNA was extracted from VLPs using the RNeasy Plant Mini Kit (QIAGEN, Hilden, Germany). Approximately 0.5 μg of extracted RNA resolved by electrophoresis on a 1.5% (w/v) denaturing formaldehyde agarose gel and transferred by capillary to a positively charged nitrocellulose membrane (Roche, Merck KGaA, Darmstadt, Germany) using SSC 10× buffer. For northern blot analysis the membranes were incubated with digoxigenin-labelled RNA probes targeting specific sequences of interest. Three different RNA probes were prepared (PVX-RdRp-probe, SPFMV-CP-probe and SPMMV-CP-probe) with the DIG starter labelling kit (Roche, Merck KGaA, Darmstadt, Germany) and using as templates PCR fragments incorporating the T7 promoter sequence for in vitro transcription. Detection was conducted using alkaline phosphatase-conjugated anti-DIG antibody and the signal was captured by a ChemiDoc$^{TM}$ imaging system (BioRad Laboratories, Hercules, USA) after applying the chemiluminescence reagent CDP-Star (Roche, Merck KGaA, Darmstadt, Germany).

**Thermal shift assays**. A method based on differential scanning fluorimetry assay[73] was used to compare the thermal stability of purified VLPs. Briefly, Sypro Orange was added as an extrinsic fluorophore[43,44] to the preparations of VLPs and readings were determined along a gradient of temperatures in a real time qPCR Mx3005p equipment (Agilent). To assure data reproducibility, duplicates for each sample were used in both assays. The melting temperatures (Tm) were calculated after data normalization using the first derivative of a native fraction for each sample to determine the maximum slopes in the different sections of the curves.

**Transmission electron microscopy (TEM) imaging**. For electron microscopy visualization of VLPs, samples (clarified crude extracts, or fractions of purified VLPs) were applied on carbon-coated copper grids (EM Resolutions, Sheffield, UK) and allowed to be absorbed for 1 min, before removing the excess with blotting paper, and later stained with 2% (w/v) of uranyl acetate for 1 more min. Images were taken using a Talos F200C TEM (Thermo Fischer Scientific, Waltham, USA) fitted with Gatan OneView camera. For routine testing of purifications performed at CRAG laboratory, a Jeol microscope JEM-1400 (120 kV) was also used. Size measurements to determine the distribution of VLP lengths was done in well contrasted images through measuring up to 150 individual particles selected unbiasedly in the image following a boustrophedon itinerary, calculating statistics parameters with Excel.

**Sample preparation for CryoEM**. CryoEM grids were prepared applying three μL of each sample to holey carbon grids (Quantifoil, 1.2/1.3), incubated briefly in the chamber at 4 C with 100% humidity, and vitrified in liquid ethane using a Vitrobot IV (Thermo Fisher Scientific). Grids were then transferred to a Titan Krios microscope, operated at 300 keV, equipped with Falcon III detector for data collection. In total, 3276 movies (SPFMV) and 2260 movies (SPMMV) were collected, each movie containing 32 frames with a dose of ~1e$^-$/A$^2$ per second.

**Image processing**. Image processing statistics are shown in Table 1. All image processing steps were carried out using RELION 3.1[74] unless stated otherwise. For both SPFMV and SPMMV datasets, movies were corrected for beam-induced motion using the RELION implementation of motioncor2[75]. The motion-corrected micrographs were estimated for contrast transfer function (CTF) parameters using CtfFind4[76].

Helical segments were picked using the helical segment picking tool in crYOLO[77] based on training a model using manually picked segments from a subset of micrographs. The automatically picked helical segment coordinated were imported into relion and were extracted in a box-size of 208 by 208 pixels. Segments were extracted with an initial estimate of 8 asymmetric units per segment. The extracted segments were analysed using several rounds of 2D classification, using 2D helical settings implemented in RELION[74]. Segments with secondary structure elements visible in the class averages were selected for further processing, and junk classes discarded. For the initial reference map, a cylinder with the diameter of the viruses was generated using relion_helix_toolbox and used as input for 3D classification with a single class as the output. Initial estimates of the helical parameters were a helical rise of 4.0 Å and a helical twist of 40°, based on previous structures of the *Potyviridae*[21–23]. The output map was then used as input for a second round of 3D classification, with five classes, running a global search for the alignment and searching for helical parameters (keeping the tilt priors fixed). One out of the five classes gave a 3D map where unambiguous separation of CP subunits was observed. This subset of particles was then refined using helical 3D refinement, searching for local helical twist and rise in order to refine initial helical parameters. The refined maps at 4.8 Å (SPFMV) and 6.5 Å (SPMMV) resolutions were then used as input for 3D classification with no alignment in order to separate particles based on the accuracy of refinement. The best refined particles selected (48,042 segments for SPFMV and 30,663 segments for SPMMV) were then refined again before running Bayesian polishing and iterative rounds of CTF refinement, resulting in improved maps with resolutions of 2.6 Å (SPFMV) and 3.0 Å (SPMMV), based on 0.143 FSC gold standard criterion. The maps were sharpened using the PostProcess programme in RELION and their local resolution assessed using Resmap[78]. The sharpened and unsharpened maps were used for model

building for the two structures. The workflow followed for image processing is summarized as supplemental material (Supplementary Fig. 7).

**Model building**. Homology models for SPFMV and SPMMV were generated using Phyre2[79] based on existing structures in the protein data bank (PDB). The CP structures were initially rigid-body fitted into the cryoEM density maps using the 'fit in map' function in UCSF Chimera[80]. Based on the preliminary fits, the models for the coat proteins were then manually refined in Coot[81], adding alanine residues in place for the WT sequence where the EM map density was of insufficient resolution. Based on the resolution within the RNA binding regions of the CP in both maps, five nucleotides of the RNA were traced de novo in Coot as a poly-U chain. Models were subjected to real space refinement in Phenix[82], together with ISOLDE[83] to fix model geometry outliers. Upon each refinement round, the models were inspected in Coot to check for any outliers or remaining clashes. The coat proteins were then symmetrised in UCSF Chimera based on the empirically determined helical parameters. In order to improve the accuracy of the model at the inter-subunit interfaces, three coat protein subunits per helical turn were selected and refined using real-space refinement in Phenix. Inter-subunit interface calculations were done using the PISA server[84] and electrostatic calculations from the fitted models were carried out using UCSF Chimera. The Phenix validation programme was run on the refined models to assess for model geometry parameters until no further improvements could be observed.

**Structural predictions using AlphaFold**. Full-length sequences for SPFMV and SPMMV CP, along with other CPs of ipomoviruses, were used for sequence-based structure prediction using AlphaFold server[85]. In brief, protein sequence files were given as input for alpha fold prediction, running a multi-sequence alignment and secondary structure prediction on its pre-trained network. Top scoring models were analysed and coloured according to the b factor or P score using UCSF ChimeraX. Figures were generated using UCSF ChimeraX.

**Statistics and reproducibility**. Transient expressions of CPs and purifications of VLPs were performed following the described protocols independently in two laboratories, JIC and CRAG, with reproducible results. For CryoEM structure determination the parameters and statistics of data collection, image analysis, and model building are shown in Table 1.

**Reporting summary**. Further information on research design is available in the Nature Portfolio Reporting Summary linked to this article.

## Data availability
The cryoEM maps were deposited to the electron microscopy databank (EMDB) with accession codes 8ACB (SPFMV) and 8ACC (SPMMV), and the models deposited to protein data bank (PDB) with accession codes of EMD-15345 (SPFMV) and EMD-15346 (SPMMV) respectively. The raw data from which these models were derived were deposited in the EMPIAR database as EMPIAR-11490 - SPMMV and EMPIAR-11491 - SPFMV. Source data for the graphs in Fig. 3 are available as Supplementary Data. Uncropped images of gels and blot are provided in Supplementary Fig. 8. Other data are available from the authors upon reasonable request.

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

## Acknowledgements

We acknowledge the Horticultural and Bio-imaging platforms at JIC, the Plant Growth core facility at CRAG, and the Servei de Microscòpia UAB whose support were invaluable throughout this study. Advice on thermal-shift by Adrián Velázquez Campoy (Instituto Universitario de Investigación de Biocomputación y Física de Sistemas Complejos, BIFI, Universidad de Zaragoza, Spain) is highly appreciated. We thank Adrian Valli (CNB-CSIC, Madrid, Spain) for helpful discussions. Funding provided by grant PID2019-105692RB-100 (to J.J.L.M.) and CEX2019-000902-S (awarded to CRAG), both from Agencia Estatal de Investigación AEI, Spain, MCIN/AEI/10.13039/501100011033, by CERCA Programme / Generalitat de Catalunya grant 2018 FI_B 00329 and EMBO Short-Term Fellowship 8695 (to O.C.), by EC Horizon 2020 project Pharma-Factory 774078, and UK Biotechnology and Biological Sciences Research Council (BBSRC) Grants BB/R001669/1 and BB/T004703/1 (to G.P.L.), and BB/T004525/1 (to N.A.R. and G.P.L.), and by Institute Strategic Programme Grant "Molecules from Nature—Enhanced Research Capacity" BBS/E/J/000PR9794 and the John Innes Foundation.

## Author contributions

Conceptualization: O.C., E.C.T., G.P.L., N.A.R. and J.J.L.M. Methodology, investigation and visualization: O.C., A.J., M.J.B., E.C.T., G.P.L., N.A.R. and J.J.L.M. Supervision: G.P.L., N.A.R. and J.J.L.M. Writing-original draft: O.C., A.J., G.P.L., N.A.R. and J.J.L.M. Writing-review & editing: O.C., A.J., M.J.B., E.C.T., G.P.L., N.A.R. and J.J.L.M.

## Competing interests

The authors declare no competing interests.
