## [Peer Review File · Communications Biology]

Reviewers' comments:

Reviewer #1 (Remarks to the Author):

The paper "... Cryo-EM structures of a potyvirus and an ipomovirus that infect sweet potato as common host" by Chase et al. describes the cryo-EM structures of two filamentous flexible plant viruses, SPFMV and SPMMV, at relatively high resolutions. The structures of the two viruses are interesting, despite not totally new, since the structures at near-atomic resolution of other three members of the same family are known. The structures have been properly determined, the paper clearly written and the results clearly presented.

The only weak part of the paper is the Discussion Section, which is too long in my opinion and could be slightly shortened. Some of the conclusions appear somehow forced, for example the potential applications of the different thermal stability of the two viruses is perhaps overestimated. The last two sentences also appear too optimistic.

In conclusion, the paper is worth being published, since it increments the number of structures at near-atomic resolution of filamentous flexible viruses, a class of viruses that cannot be crystallized and whose atomic models are becoming available in the last few years only thanks to the cryo-EM revolution.

Minor technical points:

Line 5205 and 256 ... of 2.6 Å (SPFMV) and 2.9 Å (SPMMV)... The resolution of SPMMV is 3 Å, not 2.9 (2.99 from Table S1) and the model resolution 2.7 and 3.1. I don't know if this can be defined "high-resolution" and "near-atomic". I would say "relatively high resolution" or "quite high resolution".

Line 268. ... correspond to Arg 235, Ser 172, Asp 248 (SPFMV) and Arg 262, Ser 152, Asp 236 (SPMMV). Asp 248 for SPFMV and Arg 262 and 236 for SPMMV are not listed in Table S4. Why? Perhaps a problem in the numbering system?

In supplementary Figure S5, the AlphaFold models should be superimposed (or shown close to) the structures of the corresponding CP, to allow the reader to appreciate the similarities (and the differences) and to evaluate the prediction, not only of the portion of protein not visible in the structure, but also of the visible one.

The Figures of the overall structures of the entire viruses are very small and it is difficult to appreciate the details. I hope they will be printed at a larger size.

Reviewer #2 (Remarks to the Author):

The current manuscript from Chase and co-workers delivers cryoEM structures for SPFMV and SPMMV VLPs at high resolutions. The structural data are of high quality and portrait the first structure for an ipomovirus. By the comparison of the structures the authors comment on the conservation of the helical structures and the 3D fold of the CPs for both viruses. This has been described before for representatives of the family potyviridae, and even with members of other families (alphaflexiviridae). Nevertheless, the work contributes significantly to the field by providing additional and high-quality new structures for flexible filamentous plant viruses.

There are some issues that need clarification.

Major issue

In the abstract (lines 34-36) it is said

"These allowed a direct comparison to be made between the structures of the two virus particles that revealed potential reasons for their differences in stability."

It is true that the authors present both structures and experiments that show different stability of the VLPs, but the authors do not comment anything about the structural basis for these differences, as the sentence suggests. So, this statement needs to be downplayed.

Minor issues

1. The number of micrographs used in the final cryoEM maps is the same as the initial number (Table S1). This is a bit odd. Did not the authors remove some of the pictures based on some figures of merit after motion correction and ctf estimation?
2. In the same Table S1 the symmetry is set to C1. This is partially true, but the cryoEM image processing has taken advantage (and applied) helical symmetry. This must be incorporated in the table, with some general parameters (raise and torsion) included.
3. Line 265 "two to three residues" expression is not clear. There are three conserved residues in all the reported structures for flexible filamentous plant viruses.

Reviewer #3 (Remarks to the Author):

Summary:

The manuscript describes structures (atomic models from cryoEM maps) of the core protein (CP) from two sweet potato viruses from different genera within the Potyviridae family. One is SPFMV (a Potyvirus) and the other is SPMMV (an Ipomovirus). These are both sweet potato viruses, but are transmitted by different vectors (aphids/whitefly). Both are filamentous RNA containing viruses, and the structures are determined from RNA containing virus-like particles. There are structures of Potyvirus CP in the PDB (cited in the manuscript), but no examples of an Ipomovirus.

A difference in thermal stability of the two different VLPs is measured using differential scanning fluorimetry. The core of the CP structural motifs is similar to previously determined Potyvirus structures, as is their helical arrangement. New insights provided are details on subunit contacts explaining different stability of the VLP filaments (these include the core region, and regions of the N & C terminal domains). The extreme N-terminal and C-terminal regions are not resolved in the cryoEM maps, indicating flexibility, and the analysis is supported by AlphaFold models of these parts. The N-terminal regions are implicated in cell infection and vector interactions, and analysis of the N-terminal parts of these structures indicates structural features that might play a role.

Details of the RNA binding in the RNA binding pocket are also seen and compared. The similarities in the two structures indicate a wider applied mode in the Potyviridae family, not specific to just Potyviruses.

Overall impression:

The work and presentation of results is of good quality. The manuscript is clearly written. All data and structures are released; and are available for further detailed examination by other researchers. The methods are described in adequate detail. The results will be of interest to virologists, structural virologists, and those interested in virus assembly, or protein complex assembly in general. The RNA-protein interactions and RNA packaging features are also interesting. The results could support development of antiviral strategies.

Specific comments:

1. P3, line 77: It is not clear of the impact on crops these viruses would have without co-infection of SPCSV, or if clearing them without removing SPCSV would be of benefit.
2. P8, line 250: This explanation is not clear. The point is that averaging over different conformations weakens the signal, and blurs details.
3. The title could be more engaging, e.g. hint at some of the results.

COMMSBIO-22-3479 "CryoEM structures of a potyvirus and an ipomovirus that infect sweet potato as common host"

We thank the Editor and Reviewers for their consideration of our manuscript. We are pleased to note that all three reviewers regard the work reported as being of good quality, clearly written and worthy of publication. However, they raised several specific points which we address below:

POINT-BY-POINT RESPONSE TO THE REFEREES

Reviewers' comments:

Reviewer #1 (Remarks to the Author):

The paper "... "Cryo-EM structures of a potyvirus and an ipomovirus that infect sweet potato as common host" by Chase et al. describes the cryo-EM structures of two filamentous flexible plant viruses, SPFMV and SPMMV, at relatively high resolutions. The structures of the two viruses are interesting, despite not totally new, since the structures at near-atomic resolution of other three members of the same family are known. The structures have been properly determined, the paper clearly written and the results clearly presented.

The only weak part of the paper is the Discussion Section, which is too long in my opinion and could be slightly shortened. Some of the conclusions appear somehow forced, for example the potential applications of the different thermal stability of the two viruses is perhaps overestimated. The last two sentences also appear too optimistic.

Response: Thank you for the general good impression. We have reduced the discussion to make it more readable. The wording of the final sentences has been softened.

In conclusion, the paper is worth being published, since it increments the number of structures at near-atomic resolution of filamentous flexible viruses, a class of viruses that cannot be crystallized and whose atomic models are becoming available in the last few years only thanks to the cryo-EM revolution.

Minor technical points:

Line 5205 and 256 ... of 2.6 Å (SPFMV) and 2.9 Å (SPMMV)... The resolution of SPMMV is 3 Å, not 2.9 (2.99 from Table S1) and the model resolution 2.7 and 3.1.

Response: Thanks for pointing to these inconsistencies. The values have been amended in the text.

I don't know if this can be defined "high-resolution" and "near-atomic". I would say "relatively high resolution" or "quite high resolution".

Response: We thank the reviewer for their comment about the use of 'resolution' terminology. In the field, it is common if the structures are at 3.0 Å (or sub 3 Å) to be referred to as 'high-resolution'. Understandably, it is not 'atomic' resolution but it is good enough to model the individual amino acids or nucleotides of the macromolecular complex. Therefore the maps reported in the manuscript would be considered 'high-resolution'.

Line 268. ... correspond to Arg 235, Ser 172, Asp 248 (SPFMV) and Arg 262, Ser 152, Asp 236 (SPMMV). Asp 248 for SPFMV and Arg 262 and 236 for SPMMV are not listed in Table S4. Why? Perhaps a problem in the numbering system?

Response: We thank the reviewer for picking up the inconsistency between the residues reported in the text and in the table.

Table S4 provides the reports generated by PISA analysis of structures with interactions based on a pre-established cutoff for distance-based hydrogen bonds ("PISA considers whether an H bond be present if the distance between the heavy atoms, donor and acceptor, is less than 3.89Å", according to the info at https://www.ebi.ac.uk/pdbe/pisa/pi_tips.html). It seems that the analysis missed out residues Asp248 (SPFMV) and Arg 262 and Asp 236 (SPMMV) which are present within the binding pocket, but at a calculated distance away probably exceeding the limit. This has been mentioned in a note in the supplemental table S4.

Additionally, we have updated the name of the section to better explain its contents: "Interactions between the coat protein subunits and incorporated RNAs"

In supplementary Figure S5, the AlphaFold models should be superimposed (or shown close to) the structures of the corresponding CP, to allow the reader to appreciate the similarities (and the differences) and to evaluate the prediction, not only of the portion of protein not visible in the structure, but also of the visible one.

Response: A modified version of the supplemental figure has been prepared according to the suggestions

The Figures of the overall structures of the entire viruses are very small and it is difficult to appreciate the details. I hope they will be printed at a larger size.

Response: We leave this point to the editors in order to enlarge the size of the figures in the final version for publication, or to split the figure 4 in two, if preferred.

Reviewer #2 (Remarks to the Author):

The current manuscript from Chase and co-workers delivers cryoEM structures for SPFMV and SPMMV VLPs at high resolutions. The structural data are of high quality and portrait the first structure for an ipomovirus. By the comparison of the structures the authors comment on the conservation of the helical structures and the 3D fold of the CPs for both viruses. This has been described before for representatives of the family potyviridae, and even with members of other families (alphaflexiviridae). Nevertheless, the work contributes significantly to the field by providing additional and high-quality new structures for flexible filamentous plant viruses.

There are some issues that need clarification.

Major issue

In the abstract (lines 34-36) it is said

“These allowed a direct comparison to be made between the structures of the two virus particles that revealed potential reasons for their differences in stability.”

It is true that the authors present both structures and experiments that show different stability of the VLPs, but the authors do not comment anything about the structural basis for these differences, as the sentence suggests. So, this statement needs to be downplayed.

Response: The wording of this sentence has been adapted, changing "revealed potential reasons" for "provide the basis".

Minor issues

1. The number of micrographs used in the final cryoEM maps is the same as the initial number (Table S1). This is a bit odd. Did not the authors remove some of the pictures based on some figures of merit after motion correction and ctf estimation?

Response: We thank the reviewer for their comment on Table S1. The reason for numbers staying consistent between data collection and image selection is that during our analysis, we relied on sorting the 'good vs. bad' images based on 2D classification of picked segments instead of manually curating micrographs. Both methods work but due to not manually curating micrographs (and CTF estimated files), the numbers stayed the same.

2. In the same Table S1 the symmetry is set to C1. This is partially true, but the cryoEM image processing has taken advantage (and applied) helical symmetry. This must be incorporated in the table, with some general parameters (raise and torsion) included.

Response: We thank the reviewer for their suggestion and we have amended the supplementary Table S1 accordingly.

3. Line 265 “two to three residues” expression is not clear. There are three conserved residues in all the reported structures for flexible filamentous plant viruses.

Response: We thank the reviewer for the suggestion, and we have amended the main text body accordingly.

Reviewer #3 (Remarks to the Author):

Summary:

The manuscript describes structures (atomic models from cryoEM maps) of the core protein (CP) from two sweet potato viruses from different genera within the Potyviridae family. One is SPFMV (a Potyvirus) and the other is SPMMV (an Ipomovirus). These are both sweet potato viruses, but are transmitted by different vectors (aphids/whitefly). Both are filamentous RNA containing viruses, and the structures are determined from RNA containing virus-like particles. There are structures of Potyvirus CP in the PDB (cited in the manuscript) , but no examples of an Ipomovirus.

A difference in thermal stability of the two different VLPs is measured using differential scanning fluorimetry. The core of the CP structural motifs is similar to previously determined Potyvirus structures, as is their helical arrangement. New insights provided are details on subunit contacts explaining different stability of the VLP filaments (these include the core region, and regions of the N & C terminal domains). The extreme N-terminal and C-terminal regions are not resolved in the cryoEM maps, indicating flexibility, and the analysis is supported by Alphafold models of these parts. The N-terminal regions are implicated in cell infection and vector interactions, and analysis of the N-terminal parts of these structures indicates structural features that might play a role.

Details of the RNA binding in the RNA binding pocket are also seen and compared. The similarities in the two structures indicate a wider applied mode in the Potyviridae family, not specific to just Potyviruses.

Overall impression:

The work and presentation of results is of good quality. The manuscript is clearly written.

All data and structures are released; and are available for further detailed examination by other researchers. The methods are described in adequate detail. The results will be of interest to virologists, structural virologists, and those interested in virus assembly, or protein complex assembly in general. The RNA-protein interactions and RNA packaging features are also interesting. The results could support development of antiviral strategies.

Specific comments:

1. P3, line 77: It is not clear of the impact on crops these viruses would have without co-infection of SPCSV, or if clearing them without removing SPCSV would be of benefit.

A short sentence has been incorporated for clarification of the point: "[...] and although it is often not particularly damaging as a single infection, it can be quite severe in co-infections with [...]"

2. P8, line 250: This explanation is not clear. The point is that averaging over different conformations weakens the signal, and blurs details.

We have modified the sentence to include a better explanation as suggested. The text now reads: "*...it is likely that this structural motif is averaged out during cryoEM 3D reconstruction due to the possible high flexibility of the N-terminal loop preceding it [...]*".

3. The title could be more engaging, e.g. hint at some of the results.

Thanks for the opportunity.

The journal instructions establishes a limit of 15 words or fewer, as alternative, we propose:

"CryoEM and stability analysis of virus-like particles of potyvirus and ipomovirus infecting a common host" (15 words)